# Socio-Demographic and Diet-Related Factors Associated with Insufficient Fruit and Vegetable Consumption among Adolescent Girls in Rural Communities of Southern Nepal

**DOI:** 10.3390/ijerph16122145

**Published:** 2019-06-17

**Authors:** Jitendra Kumar Singh, Dilaram Acharya, Salila Gautam, Mandira Adhikari, Ji-Hyuk Park, Seok-Ju Yoo, Kwan Lee

**Affiliations:** 1Department of Community Medicine, Janaki Medical College, Tribhuvan University, Janakpur 456000, Nepal; 2Department of Preventive Medicine, College of Medicine, Dongguk University, Gyeongju 38066, Korea; dilaramacharya123@gmail.com (D.A.); skeyd@naver.com (J.-H.P.); kwaniya@dongguk.ac.kr (K.L.); 3Department of Community Medicine, Kathmandu University, Devdaha Medical College and Research Institute, Rupandehi 32907, Nepal; 4Department of Public Health, Sanjeevani College of Medical Sciences, Butwal, Rupandehi 32907, Nepal; Salilagautamacharya@gmail.com; 5Nepal Development Society, Bharatpur 44200, Nepal; adhikarimandira2013@gmail.com

**Keywords:** adolescent girls, dietary factors, fruit and vegetable consumption, Nepal

## Abstract

Sufficient fruit and vegetable (FV) consumption has been associated with reduced risks of chronic diseases and adverse health conditions. However, the determinants of insufficient of FV intake among adolescent girls in Nepal have not been determined. This study was undertaken to identify associations between socio-demographic and diet-related factors with insufficient fruit and vegetable consumption among adolescent girls living in rural communities. This community-based, cross-sectional study was conducted on 407 adolescent girls from rural communities in the Bateshwar rural municipality of Dhanusha district, Southern Nepal between 12 October, 2018 and 14 December, 2018. The study subjects responded to FV consumption and dietary factor-related questionnaires, and anthropometric measurements were taken. Data were analyzed using the univariate logistic regression followed by multivariable logistic regression analyses. Unadjusted and adjusted odds ratios with 95% confidence intervals (CI) are reported. From the 407 study subjects, 359 (88.2%) reported insufficient FV consumption. The factors significantly associated with insufficient FV consumption were education to under the 10th grade, household income in the first tercile, lack of awareness of the importance of FV consumption, the non-availability of FVs at the household level, the low level of dietary diversity, and undernutrition (BMI (body mass index) (<18.5)). The study shows almost 90% of adolescent girls consumed inadequate amounts of FV and that socio-demographic and dietary factors should be taken into account while designing preventive strategies to increase fruit and vegetable consumption to recommended levels.

## 1. Introduction

An expert committee of the World Health Organization (WHO) and the Food and Agriculture Organization (FAO) has recommended a dietary intake of at least 400 g of fruits and vegetables (FVs) per day (roughly equivalent to five servings per day) [1]. The presence of antioxidants and phytochemicals in FVs promote health by neutralizing free radicals [2,3,4]. The findings from a wide range of prospective studies [5], systematic reviews, and meta-analyses [6,7,8], suggest that adequate FVs consumption can protect against the development of breast, hepatic and lung cancer. Diets rich in FVs can also help prevent several other non-communicable diseases and conditions, such as type 2 diabetes, metabolic syndrome [1,9,10,11,12], cardiovascular diseases [13,14] and lower all-cause mortality and increase longevity [14,15]. In addition, a reduced amount of FVs consumption is a major risk factor of obesity among adolescents and young adults [16].

It is important for children and adolescents to consume the right quantity and quality of FVs because the dietary habits are formed during transition from children to adolescents. Thus, the adoption of inappropriate dietary habits from early age has far reaching health consequences [17,18]. Insufficient consumption of recommended levels of FVs has been consistently reported for adolescents in developing and developed countries despite the numerous advantages [19,20,21]. The latest study on the topic reported that fewer than 30% of adolescents in 49 low-and-middle income countries (LMICs) consumed FVs at WHO recommended levels in 2018 [19]. In a study conducted in five South Asian countries (India, Indonesia, Myanmar, Sri Lanka, and Thailand), the FVs consumption of 76.3% for 13 to 15 year-olds was found to be inadequate [20]. In Nepal, a recent nationwide STEPwise approach to Surveillance (STEPS) Survey revealed that almost all (99%, 95% CI: 97.8–99.5) 15 to 29 year old consumed less than five servings of FVs per day on average [22]. In addition, local area research has reported 98% of a peri-urban adult population [23], 85.6% of undergraduate medical students [24], 96.6% of adults living in a mountainous region [25], 76% of 18–29 year old living in Kathmandu [26], consumed insufficient quantities of FVs. However, FVs consumption by adolescent girls living in rural communities is limited, although, a recent study on rural communities of Nepal concluded that about 87% pregnant women did not consume the recommended levels of FVs [27].

A large number of studies have sought to identify factors associated with insufficient FVs consumption among adolescents in LMICs, and those identified include personal characteristics—age; gender; family income; education level; education level of the family head [7,28,29,30]; lifestyle-related factors such as smoking, alcohol consumption, sedentary behavior, soft drink consumption; insufficient physical activity [28,29,30]; and overweightness and obesity [30]. Moreover, a recent systematic review identified 85 unique determinants and highlighted the importance of the effects of race/ethnicity, FVs preferences, and maternal FVs intake on FVs consumption among low-income youths less than 20 years old [31].

Given adolescence is an important period of life in terms of shaping health-promoting behavior, the consumption of recommended amounts of FVs is dependent on many factors [17,18,19,32]. In addition, historically the nutrition status of girls and women is poor for dietary diversity as compared with boys [33]. Girls are neglected, but their nutrition status is important, not only for them, but also for intergenerational effects on nutrition outcomes. Therefore, the authors decided to quantify to what extent adolescent Nepalese girls living in rural communities are consuming sufficient amount of FVs and to communicate the risk factors associated with inadequate consumption, in the hope that the information gleaned might aid the design of evidence-based health promotion and preventive strategies. Accordingly, this study aimed to identify the associations of socio-demographic and diet-related factors with insufficient fruit and vegetable consumption among adolescent girls living in rural communities in southern Nepal.

## 2. Materials and Methods

### 2.1. Study Design and Setting

This community-based, cross-sectional study was conducted in rural communities of Bateshwar rural municipality in Dhanusha district, southern Nepal between 12 October, 2018 and 14 December, 2018. The study area is located in southern Terai of province number two in Nepal. The rural municipalities (also called Gaunpalika) are third level administrative units in rural Nepal. Bateshwar rural municipality has a population of 21,530 [34] and the total population of Dhanusha district is 754,777, which includes 84,860 adolescent (10 to 19 years of age) girls [35,36]. The study area is predominantly inhabited by rural communities, and thus, most residents perform agricultural work, although during recent years, foreign company employment has become more popular among skilled and unskilled manual workers [36]. 

### 2.2. Sample Size and Sampling Procedure

A multistage random sampling procedure was used to select study subjects. First, Bateshwar rural municipality was selected from among 6 rural municipalities in the Dhanusha district using a random number table. Second, two of 5 wards in Bateshwar with approximately equal populations and household numbers were also selected using a random number table. Third, a complete list of households and individuals in both wards was obtained from the office of the Baleshwar rural municipality. Fourth, households were systematically selected and an adolescent girl (10–19 years) was selected from appropriate households for interview. When there were two or more adolescent girls in one household, all were considered. Married adolescent girls and those with a physical or mental health issue were excluded. 

OpenEpi software was used to calculate the sample size [37]. This calculation was based on a report that the percentage of Nepalese adolescents that consume the recommended five or more servings of FVs per day was approximately 4% [38]. Assuming a 2.5% marginal error at a 95% confidence level and considering a design effect of 1.5 to account for the intra-cluster effect, 354 adolescent girls were required. Based on a presumed non-response rate of 20%, 425 adolescent girls were invited to participate in the survey. However, 18 did not respond to the invitation, and thus, 407 were included in the study (a response rate of 95.7%).

### 2.3. Data Collection and Anthromometric Measurements

Data were collected during face-to-face interviews by trained researchers using a structured questionnaire adapted from the Nepalese Adolescent Nutrition Survey (2014), the Food Frequency Questionnaire (FFQ), and the dietary diversity questionnaire [38,39,40]; all of which were of already piloted study instruments. Interviews were followed by measuring anthropomorphic variables using calibrated instruments. The survey questionnaires consisted of three parts: (i) Personal profiles (socio-demographic characteristics); (ii) fruit and vegetable consumption, dietary behavior/diet-related factors, and anthropometric measurements; and (iii) the Food Frequency Questionnaire (FFQ). The FFQ was used to assess the frequency and amount of FV consumption and the dietary diversity questionnaire was used to assess dietary diversity [39,40]. Anthropometric measurements, such as weight and height, were obtained using standardized and calibrated study instruments. Body weights were measured to the nearest 100 g using calibrated portable scales with subjects wearing light clothing but without footwear. Heights were measured to the nearest centimeter using a calibrated measuring rod in a full standing position without footwear. Body mass indices (BMIs) were calculated by dividing weight in kilograms by height in meters squared. The questionnaires were translated into the local language (Nepali) and then back-translated into English to ensure translations were accurate. 

### 2.4. Definitions of Variables

The main study outcome variable was sufficient intake of FVs and was defined as sufficient if a subject consumed five portions (400 g) or more per day. Based on the available literature, one cup (250 mL) of raw green leafy vegetables or a half cup (125 mL) of cooked or chopped raw vegetables was considered equivalent to one serving of vegetables. Similarly, one portion of fruits was defined as one whole medium-sized fruit (e.g., apple, banana or orange) or half a cup of chopped, cooked, canned fruit, or fruit juice [39]. The study used show cards to collect information for the FFQ on FV intakes. 

The independent study variables were: (i) Socio-demographic variables (age, caste/ethnicity, religion, household structure, education, main household occupation, household income); (ii) dietary-related factors; (iii) dietary diversity; and (iv) nutritional status. The subjects were dichotomized as early (10–14 years) or late adolescents (15–19 years). The castes/ethnicities were classified as: (i) Upper caste group—Brahmin, Chhetri, and other relatively-advantaged Terai caste groups (Yadav, Shah, Koiri, Thakur); (ii) Adibasi/Janajati–Janajati or indigenous groups; and (iii) Dalit (relatively disadvantaged) [41]. Religions were classified as Hindu or others (Muslim/ Buddhists, Christians). Household structures were classified as (i) nuclear or (ii) joint or extended. Education was classified as: (i) <10th grade (less than 10 years of schooling) or (ii) ≥10 grade. The main household occupations (parenteral occupations) were categorized as: (i) Agricultural work on farms owned by the household; (ii) services/business work in private companies for the government or their own business; (iii) foreign employment when working in a foreign country; and (iv) others, which included skilled, unskilled, or daily work. Household incomes per month were recorded in Nepali rupees and categorized using terciles: (i) 1st tercile (income <11,243 (Nepalese Rupees (NRs)/month)); (ii) 2nd tercile (11,243–22,343 NRs/month); and (iii) 3rd tercile (>22,343 NRs/month). The subject-reported FV consumptions were in accordance with World Health Organization (WHO) recommendations for FV consumption [42]. Initially, subjects were asked about FV requirements per day and responses were categorized as yes (correct response) or no (incorrect response). The availabilities of FVs at the household level were classified as: (i) Yes (if grown on the farm or afforded them); or (ii) no. Dietary habits were classified as vegetarian or non-vegetarian. Meal frequencies were classified as <4 or ≥4 per day.

The consumption of any amount junk food or processed food (such as chips, Kurkure, lays, noodles, salty cookies, biscuits, cakes etc.) at least once a week was classified as yes or no [38]. Addictions to alcohol, cigarettes, gutka, pan masala, betel-quid, khaini or surti, zarda, snuff, and gul were also recorded as yes or no as previously reported [41]. The dietary diversity was classified as low, medium, or high based on food items consumed in 16 food groups as recommended by the FAO. The consumption of food in ≤ 3 food groups was considered as low, 4–5 food groups as medium, and ≥6 food groups as high dietary diversity [40,43]. In order to assess nutritional statuses, Body Mass Indices (BMIs) were calculated using measured heights and weights as described above. The study participants were classified as underweight (BMI < 18.5), normal (BMI between 18.5 and 24.9), overweight (BMI between 25.0 and 29.9), or obese (BMI ≥ 30) [44], but overweight and obese were merged into one category (overweight/obese BMI ≥ 25).

### 2.5. Ethical Considerations

The study protocol was approved beforehand by the Ethical Committee of Janaki Medical College, Tribhuvan University, Nepal (approval number: 16-2075-076). The study purpose and procedure were explained to each participant before data collection was started or before anthropomorphic measurements were taken. All study subjects provided written informed consent. In addition, written parenteral consent was received for each study subject who were below 18 years of age. Privacy and confidentiality were fully maintained throughout and personal identifiers were removed prior to data analysis.

### 2.6. Statistical Analysis

Univariable logistic regression was used to assess associations between FV intake and independent variables of interest, and all factors found to be significant (*p* < 0.05) were included in the multivariable logistic regression analysis with backward elimination to control for confounders. This study used Hosmer-Lemeshow goodness-of-fit to examine model fitness. The unadjusted and adjusted odds ratios with 95% confidence intervals (CI) are reported. The analysis was performed using the Statistical Package for Social Sciences version 22.0 (SPSS, IBM, Armonk, NY, USA). 

## 3. Results

### 3.1. Personal Profiles of the Study Subjects 

The socio-demographic characteristics are detailed in Table 1. Regarding the 407 study subjects, 359 (88.2%) reported insufficient FV consumption. The majority of the girls (61.4%) were in early adolescence, slightly more than half (56.0%) of them were from the upper caste group, and almost all (93.4%) were Hindus. The majority (69.8%) of the study participants completed <10th grade education, and 60.1% lived in a nuclear family. Nearly half (48.4%) of the households were in the service/business or agriculture occupation and slightly less than half (45.4%) had a household income in the 1st tercile. Subject educations, household types, main parenteral occupations, and household incomes were significantly associated with FV consumption (Table 1).

### 3.2. Associations between Awareness of Recommended Intake of FV, Dietary-Related Factors, and Nutrition Status with Fruit and Vegetable Consumption 

Table 2 shows the result of the logistic regression analysis with crude odds ratio which demonstrates the associations between awareness, dietary-related factors, and nutritional status and FV consumption. Adolescent girls unaware of FV consumption recommendations, with the non-availability of FVs at the household level, the low and medium level of dietary diversity, and an underweight nutritional status were found to be significantly associated with insufficient FV intake.

### 3.3. Associations between Socio-Demographic Variables, Awareness, and Dietary Factors with Fruit and Vegetable Consumption

The results of the final multivariable logistic regression analysis are presented in Table 3. Education level, household income, awareness of recommended FVs intake, availability of FVs, dietary diversity and body mass index were significant determinants of FV consumption. Adolescent girls who were educated with <10th grade of schooling were more likely (adjusted odds ratio (aOR) 2.5; 95% CI (confidence interval) (1.06–5.78)) to consume insufficient fruit and vegetables than their counterparts of ≥10th grade of schooling. Similarly, household income in the first tercile (aOR 3.9; 95% CI (1.22–12.67)), those who were unaware of the recommended fruit and vegetable consumption (aOR 3.1; 95% CI (1.20–8.23)), non-availability of FVs at household level (aOR 3.0; 95% CI (1.16–7.60)), low level of dietary diversity (aOR 2.3; 95% CI (1.04–4.97)), undernourished (BMI (<18.5)) (aOR 8.2; 95% CI (1.32–47.45)) had higher odds of insufficient FVs intake than their counterparts.

## 4. Discussion

The current study shows that the majority of adolescent girls living in rural communities in Southern Nepal consume less than the recommended amounts of fruit and vegetables. Only 11.8% of the study subjects were found to consume adequate amounts. This finding was higher than those reported in other Nepalese studies, including the nationwide STEPS survey, which reported almost all adolescent girls failed to meet FV intake recommendations [22,23,25]. This disagreement may be due to the subject age and settings differences. For example, these other Nepalese studies included adults and study settings varied. In addition, our younger study subjects may have had greater access to FVs, and fewer barriers to FV consumption. This could be explained in a way that younger age group subjects in our study setting might have to depend more on household foods than older age groups who might have preference on pre-packaged and fast foods outside leading to decreased FV consumption. On the other hand, our findings are similar to some other studies conducted on undergraduate medical students and on 18–29 year olds in Kathmandu [24,26], which suggest age and study setting may influence FV intake. Furthermore, our study participants were adolescent girls, and gender may also have influenced FV consumption as compared with studies conducted on males and females [19]. Moreover, other studies [45,46] reported that, boys did not like to eat FV as much as girls did. 

Our observations concur with previous findings [18,47,48,49] that FV consumption is positively associated with socioeconomic status of study participants and their families. This study found those being educated at under the 10th grade level and those with a household income in the first tercile were at greater risk of inadequate FV consumption, which was entirely expected as these individuals have less access and are less likely to be health conscious than those with higher incomes and levels of education [50]. In addition, those that were unaware of FV consumption recommendations and with limited access FVs at the household level were at higher risk of inadequate FV consumption, which is in line with the findings of previously published papers [18,51]. In fact, socio-economic position, affordability, individual preferences, perception, knowledge, awareness of FV intake recommendations, and home availability/accessibility are among the factors previously reported to importantly determine FV intake [18,52,53,54,55,56].

Dietary diversity and undernutrition were two other diet-related factors found to be strongly associated the risk of inadequate FV consumption. It is well known that dietary diversity provides the opportunity to consume more types of food items [57,58]. For example, Faber et al. reported that dietary diversity significantly increased the consumption of vitamin A rich FVs [58]. In contrast to a previous study [51], this study found that undernourished subjects were more likely to consume inadequate amounts of FVs, though they probably also ate less than the recommended levels of other essential nutrients. 

This study has some strengths that are worthy of mention. First, it is the first study undertaken to identify factors associated with insufficient fruit and vegetable consumption among adolescent girls. Second, the study had a very high response rate and made use of already piloted study instruments [38,39,40]. However, it also has several limitations. First, because of its cross-sectional design, this study could not establish cause and effect relationships. Second, the study had a relatively small sample size, which threatens the generalizability of our study findings to the national level. Third, the self-reporting method used might have introduced bias. Finally, this study did not include adolescent boys, which may have different findings. Further studies are needed to differentiate the gender influence on the factors affecting recommended FVs intake. 

## 5. Implications of the Study

The findings of this study can serve as a reference to further research in Nepal to explore the reasons for not consuming the recommended FVs among adolescents. In addition, it could be useful to design nutrition intervention programs targeting adolescents. There is a lack of dietary guidelines at a national level to educate and provide counseling to adolescents on dietary requirements. This study indicates the need for dietary guidelines for promoting healthy eating practices.

## 6. Conclusions

In the present study, it was found that the majority of adolescent Nepalese girls from a rural background consumed inadequate amounts of FVs. Socioeconomic and diet-related factors, such as schooling to under the 10th grade level, household incomes in the first tercile, the lack of awareness of FV consumption recommendations, the non-availability of FVs at the household level, a low level of dietary diversity, and undernutrition were observed to be significantly associated with inadequate FV consumption. We suggest these factors be taken into account while designing preventive strategies to increase fruit and vegetable intakes. Further, longitudinal studies should be performed to determine risk differences between adolescent girls and boys in rural communities in Nepal. 

## Figures and Tables

**Table 1 ijerph-16-02145-t001:** Socio-demographic factors associated with fruit and vegetable consumption among adolescent girls in rural communities in the Dhanusha district of Nepal, 2018.

Variables	Total	Fruit & Vegetable Consumption			*p*-Value
	*n* = 407, (%)	Insufficient *n* = 359 (%)	Sufficient *n* = 48 (%)	OR (95%CI)	
Age (years)					
10–14	250 (61.4)	225 (90.0)	25 (10.0)	1.5 (0.84–2.83)	0.157
15–19	157 (38.6)	134 (85.4)	23 (14.6)	Reference	
Caste/ethnicity					
Dalit	55 (13.5)	51 (92.7)	4 (7.3)	2.23 (0.75–6.58)	0.145
Aadibasi/Janajati	124 (30.5)	114 (91.9)	10 (8.1)	1.99 (0.95–4.19)	0.068
Upper caste group	228 (56.0)	194 (85.1)	34 (14.9)	Reference	
Religion					
Hindu	380 (93.4)	336 (88.4)	44 (11.6)	1.32 (0.43–4.01)	0.614
Muslim/others	27 (6.6)	23 (85.2)	4 (14.8)	Reference	
Education					
<10 grade	284 (69.8)	263 (92.6)	21 (7.4)	3.52 (1.90–6.52)	0.001
≥10 grade	123 (30.2)	96 (78.0)	27 (22.0)	Reference	
Household structure					
Joint/extended	162 (39.9)	150 (92.6)	12 (7.4)	2.15 (1.08–4.27)	0.026
Nuclear	245 (60.1)	209 (85.3)	36 (14.7)	Reference	
Main occupation in the household (parenteral)					
Others (skilled, unskilled or daily wage labor)	135 (33.2)	126 (93.3)	9 (6.7)	3.44 (1.44–8.22)	0.005
Foreign employment	75 (18.4)	66 (88.0)	9 (12.0)	1.80 (0.74–4.37)	0.191
Service/business	116 (28.5)	102 (87.9)	14 (12.1)	1.79 (0.82–3.91)	0.143
Agriculture	81 (19.9)	65 (80.2)	16 (19.8)	Reference	
Household income (in tercile)					
1st tercile	185 (45.4)	170 (91.1)	15 (8.1)	2.941.43 (6.05-)	0.003
2nd tercile	125 (30.8)	112 (89.6)	13(10.4)	2.23 (1.05–4.76)	0.037
3rd tercile	97 (23.8)	77 (79.4)	20 (20.6)	Reference	

OR: odds ratio; CI: confidence interval.

**Table 2 ijerph-16-02145-t002:** Awareness, dietary-related factors and nutritional status of adolescent girls associated with fruit and vegetable consumption in rural communities in the Dhanusha district of Nepal, 2018.

Variables	Total	Fruit & Vegetable Consumption			*p*-Value
	*n* = 407, (%)	Insufficient, *n* = 359 (%)	Sufficient *n* = 48 (%)	OR (95%CI)	
Awareness regarding fruit & vegetable consumption					
Yes	219 (53.8)	179 (81.7)	40 (18.3)	Reference	
No	188 (46.2)	180 (95.7)	8 (4.3)	5.02 (2.28–11.04)	0.001
Availability of fruits and Vegetables at household level					
Yes	252 (61.9)	211 (83.7)	41 (16.3)	Reference	
No	155 (38.1)	148 (95.5)	7 (4.5)	4.10 (1.79–9.40)	0.001
Dietary habits					
Vegetarian	76 (18.7)	64 (84.2)	12 (15.8)	Reference	
Non-vegetarian	331 (81.3)	295 (89.1)	36 (10.9)	1.53 (0.75–3.11)	0.234
Meal consumption per day					
<4 times	249 (61.2)	224 (90.0)	25 (10.0)	1.52 (0.83–2.79)	0.171
≥4 times	158 (38.8)	135 (85.4)	23 (14.6)	Reference	
Dietary diversity					
High	88 (21.6)	69 (78.4)	19 (21.6)	Reference	
Medium	259 (63.7)	233 (90.0)	26 (10.0)	5.23 (1.47–18.57)	0.010
Low	60 (14.7)	57 (95.0)	3 (5.0)	2.46 (1.28–4.72)	0.006
Junk food/processed food consumption once per week					
No	67 (16.5)	58 (86.6)	9 (13.4)	Reference	
Yes	340 (83.5)	301 (88.5)	39 (11.5)	1.19 (0.55–2.60)	0.649
Any addiction					
Yes	38 (9.3)	34 (89.5)	4 (10.5)	1.15 (0.39–3.39)	0.799
No	369 (90.7)	325 (88.1)	44 (11.9)	Reference	
Nutrition status; BMI (kg/m^2^)					
Underweight (<18.5)	99 (24.3)	97 (98.0)	2 (2.0)	9.09 (2.15–38.43)	0.003
Normal (18.5–24.9)	247 (60.7)	208 (84.2)	39 (15.8)	Reference	
Overweight/obese (≥25)	61 (15.0)	55 (88.5)	7 (11.5)	1.44 (0.61–3.41)	0.399

OR; odds ratio, BMI; body mass index.

**Table 3 ijerph-16-02145-t003:** Multivariable logistic regression analysis of socio-demographic, awareness, and dietary factors of adolescent girls with respect to fruit and vegetable consumption in rural communities in the Dhanusha district of Nepal, 2018.

Variables	aOR (95%CI)	*p*-Value
Education		
<10 grade	2.5 (1.06–5.78)	0.035
≥10 grade	Reference	
Household income (tercile)		
1st tercile	3.9 (1.22–12.67)	0.022
2nd tercile	1.4 (0.56–3.30)	0.492
3rd tercile	Reference	
Awareness regarding fruits & vegetables consumption		
Yes	Reference	
No	3.1 (1.20–8.23)	0.020
Availability of fruits and Vegetables at household level		
Yes	Reference	
No	3.0 (1.16–7.60)	0.023
Dietary diversity		
High	Reference	
Medium	1.5 (0.26–8.19)	0.668
Low	2.3 (1.04–4.97)	0.041
Nutrition status; BMI (kg/m2)		
Underweight (<18.5)	8.2 (1.32–47.45)	0.024
Overweight/obese (≥25)	2.7 (0.36–20.01)	0.909
Normal (18.5–24.9)	Reference	

All variables with a *p*-value of ≤0.05 by univariate analysis were entered into the final multivariable logistic regression model. Statistical significance was accepted for *p*-values of <0.05. aOR: adjusted odds ratio.

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
