# Peer review of "Socio-Demographic and Diet-Related Factors Associated with Insufficient Fruit and Vegetable Consumption among Adolescent Girls in Rural Communities of Southern Nepal"

_ijerph, 2019, doi:10.3390/ijerph16122145_

Round 1

Reviewer 1 Report

This study to describe associations between socio-economic and diet-related factors with insufficient fruit and vegetable consumption among adolescent girls living in rural communities.

This is a very interesting issue due to important consequences on the public health.
I found the paper well written and worthy to be published in
IJERPH after the following minor correction:
- Paragraph 3.3 Lines 210-221. This is a very long sentence, a bit confusing, please simplify it.

Author Response

Reviewer: first

 Round: first

Manuscript ID: ijerph-505147

Manuscript title: Socio-demographic and diet-related factors associated with insufficient fruit and vegetable consumption among adolescent girls in rural communities of Southern Nepal

Comments and Suggestions for Authors

General comments: This study to describe associations between socio-economic and diet-related factors with insufficient fruit and vegetable consumption among adolescent girls living in rural communities. This is a very interesting issue due to important consequences on the public health. I found the paper well written and worthy to be published in 
IJERPH after the following minor correction:

Response: Thank you very much for the encouragement. We have revised the manuscript in a number of spaces based on reviewers’ comments and the revised contents have been marked with blue colored writings to allow the reviewers’ verifications.

Comment:  Paragraph 3.3 Lines 210-221. This is a very long sentence, a bit confusing, please simplify it.

Response: Agree. Sentences have been re-structured.

Reviewer 2 Report

A cross-sectional study was conducted to examine the socio-demographic and diet-related factors associated with insufficient fruit and vegetable consumption among adolescent girls in rural communities of South Nepal. The study topic is important and have potential to inform public health policy for this venerable population group. However, the introduction was not well-structured. Moreover, the discussion should elaborate more on the implications and further research. The manuscript would also benefit from editing for English.

Here are some specific suggestions:

Title

Line 2: “Socio-economic” was used in the title and also in abstract line 31, however, the study also investigated other demographic factors. If the authors mean to express socio-economic and demographic factors, they should use “Socio-demographic”. Please use the two words carefully throughout the manuscript.

Abstract

Line 36: Multivariable logistic regression analysis was used in this study, not “multivariate”

Introduction

Line 47-50: Sentence too long and clumsy. The importance of addressing fruit and vegetable consumption in children should be moved to the second paragraph.

Line 52-61: The benefits of adequate fruit and vegetable intake and its association with diseases were mentioned here. This part could be shortened.

Line 64: “the situation is more serious in LMICs”, this sentence is not a valid conclusion from the literature cited.

Line 68-70: “Furthermore, a recent narrative review…” these few sentences could not be linked to the next paragraph. Authors could move it to the first paragraph

Line 77: “astonishingly” is used here. Given the high prevalence of insufficient addressing fruit and vegetable consumption in Nepal (from the cited studies), the high prevalence in pregnant women should not be a surprise?

Line 90: “to communicate the incremental risks associated with inadequate consumption” the paper did not address this point later on.

Line 93: “socioeconomic”: please refer to the first comment under Title.

Line 94: “fruit and vegetable”: The acronyms “FV” is used in Introduction. Please use acronyms consistently throughout the manuscript.

Why the authors only focused on adolescent girls but not include the boys?

Materials and Methods

Line 127:  Delete “socio-economic”

Line 127: fruit and vegetable consumption: Does this mean FFQ? Or a different questionnaire?

Line 145: Delete “socio-economic”

Line 160: “food and vegetable consumption”: Do you mean fruit? and vegetable consumption. Again, please use acronyms FV consistently.

Line 163: Mean frequencies: Do you mean Meal frequencies?

Line 165: Consumption of junk food or processed food was classified as yes or no. Could you elaborate on the amount of consumption? “No” means the subjects didn’t consume any junk or processed food? In the past month or year etc?

Line 176: Did the study ask for parents’ consent for adolescents under age of 18?

Result

Line 194 &198: It appears that univariate logistic regression analysis was used instead of Chi-square test as mentioned in Line 182, please revised accordingly.

Line 207 -221: Please correct the grammatical errors.

Line 219, The result on medium level of dietary diversity was not significant in Table 3.

Line 221: The result on overweight/ obese was not significant in Table 3.

The goodness fit of multivariable logistic regression should be reported.

Discussion

Line 225: 11.8% of participants were found to consume five servings of fruit and vegetables in this study. This result was not inconsistent with other Nepalese studies. The authors could say this result is somewhat higher than those reported in other Nepalese studies.

Line 230: “However” could be replaced by “On the other hand”.

Line 239: add reference after levels of “education”.

Line 249: “foods and vegetables”: Do you mean fruits and vegetables?

The discussion should elaborate more on the implications of the findings and further research. The limitations of not including boys were not mentioned

Author Response

Reviewer: Second

Round: first

Manuscript ID: ijerph-505147

Manuscript title: Socio-demographic and diet-related factors associated with insufficient fruit and vegetable consumption among adolescent girls in rural communities of Southern Nepal

Comments and Suggestions for Authors

General comments: A cross-sectional study was conducted to examine the socio-demographic and diet-related factors associated with insufficient fruit and vegetable consumption among adolescent girls in rural communities of South Nepal. The study topic is important and has the potential to inform public health policy for this vulnerable population group. However, the introduction was not well-structured. Moreover, the discussion should elaborate more on the implications and further research. The manuscript would also benefit from editing for English.

Response: Thank you very much for your encouragement. We highly appreciate the reviewer’s every comment that helped us a lot to improve our manuscript. We have revised the manuscript in a number of spaces based on reviewers’ comments and the revised contents have been marked with blue colored writings to allow the reviewers’ verifications.

 Here are some specific suggestions:

Comment: Title Line 2: “Socio-economic” was used in the title and also in abstract line 31, however, the study also investigated other demographic factors. If the authors mean to express socio-economic and demographic factors, they should use “Socio-demographic”. Please use the two words carefully throughout the manuscript.

Response: Agree. Checked and revised throughout the manuscript.

 Abstract

Comment: Line 36: Multivariable logistic regression analysis was used in this study, not “multivariate”

Response: Agree. Corrected.

 Introduction

 Comment: Line 47-50: Sentence too long and clumsy. The importance of addressing fruit and vegetable consumption in children should be moved to the second paragraph.

Response: Agree. The sentence is re-structured and moved to the second paragraph.

Comment: Line 52-61: The benefits of adequate fruit and vegetable intake and its association with diseases were mentioned here. This part could be shortened. 

Response: Agree. Revised.

Comment: Line 64: “the situation is more serious in LMICs”, this sentence is not a valid conclusion from the literature cited. 

Response: Agree. Revised.

Comment: Line 68-70: “Furthermore, a recent narrative reviews…” these few sentences could not be linked to the next paragraph. Authors could move it to the first paragraph.

Response: Agree. Revised as suggested.

Comment: Line 77: “astonishingly” is used here. Given the high prevalence of insufficient addressing fruit and vegetable consumption in Nepal (from the cited studies), the high prevalence in pregnant women should not be a surprise?

Response: Agree. Revised.

Comment: Line 90: “to communicate the incremental risks associated with inadequate consumption” the paper did not address this point later on.

Response: Agree.  Revised.

Comment: Line 93: “socioeconomic”: please refer to the first comment under Title.

Response: Agree. Corrected.

Comment: Line 94: “fruit and vegetable”: The acronyms “FV” is used in Introduction. Please use acronyms consistently throughout the manuscript.

Response: Agree. Checked and revised.

Comment: Why the authors only focused on adolescent girls but not include the boys?

Response: Agree. This issue has been addressed as one of the limitations of this study.

 Materials and Methods

Comment: Line 127:  Delete “socio-economic” 

Response: Agree. Deleted.

Comment: Line 127: fruit and vegetable consumption: Does this mean FFQ? Or a different questionnaire?

Response: Agree. Comment addressed.

Comment: Line 145: Delete “socio-economic”

Response: Agree. Deleted.

Comment: Line 160: “food and vegetable consumption”: Do you mean fruit? and vegetable consumption. Again, please use acronyms FV consistently. 

Response: Agree. Get corrected.

Comment: Line 163: Mean frequencies: Do you mean Meal frequencies?

Response: Agree. Yes, it should be meal frequencies. Corrected.

Comment:  Line 165: Consumption of junk food or processed food was classified as yes or no. Could you elaborate on the amount of consumption? “No” means the subjects didn’t consume any junk or processed food? In the past month or year etc?

Response: Agree. Revised.

Comment: Line 176: Did the study ask for parents’ consent for adolescents under age of 18?

Response: Agree. We had received the parenteral consents for each study subjects who were below 18 years of age. The same has been mentioned in this revised version of the manuscript.

 Result 

Comment: Line 194 &198: It appears that univariate logistic regression analysis was used instead of Chi-square test as mentioned in Line 182, please revised accordingly.

Response: Agree. Revised as suggested.

 Comment: Line 207 -221: Please correct the grammatical errors.

Response: Agree. Revised.

Comment:  Line 219, The result on medium level of dietary diversity was not significant in Table 3.

Response: Agree. Information about goodness fit of multivariable logistic regression reported will address this issue.

Comment:  Line 221: The result on overweight/ obese was not significant in Table 3.

Response: Agree. Information about goodness fit of multivariable logistic regression reported will address this issue.

Comment: The goodness fit of multivariable logistic regression should be reported.

Response: Agree. Honestly, Hosmer-Lemeshow test was used to test the goodness-of-fit of multivariable logistic regression model. The test do not raise any issue regarding the fitness of the model (p=0.638). The goodness fit of multivariable logistic regression model is reported in the manuscript under the heading statistical analysis in this revised version of the manuscript.

 Discussion 

Comment: Line 225: 11.8% of participants were found to consume five servings of fruit and vegetables in this study. This result was not inconsistent with other Nepalese studies. The authors could say this result is somewhat higher than those reported in other Nepalese studies.

Response: Agree. Revised.

Comment:  Line 230: “However” could be replaced by “On the other hand”.

Response: Agree. Revised.

 Comment: Line 239: add the reference after levels of “education”.

Response: Agree. Revised.

Comment: Line 249: “foods and vegetables”: Do you mean fruits and vegetables?

Response: Agree. Revised.

Comment: The discussion should elaborate more on the implications of the findings and further research. The limitations of not including boys were not mentioned

Response: Agree. Thank you very much for such a precious comment. We have now added the study implications in the discussion. One more study limitation stating the adolescent boys were excluded has been mentioned in the current version of the manuscript.

Reviewer 3 Report

Dear Editor,

Thank you for the opportunity to review the manuscript, Socio-economic and diet-related factors associated with insufficient fruit and vegetable consumption among adolescent girls in rural communities of Southern Nepal. The manuscript examines factors associated with FV consumptions among adolescent girls in rural Nepal, and has potentials to highlight key determinants of FV consumption in rural communities if authors could clarify the following parts.

1.      Introduction

-It would be helpful for the international audience to know why adolescent girls in Dhanusha district was selected to be an interesting case study to identify factors associated with FV consumption. For example, why only adolescent girls were selected? Why adolescent boys were not included? How is Dhausha district different from Singhuli district studied by Dhungana et al (2014)?

-“STEPS survey” in line 71 should be defined in parenthesis the first time appeared in main text.

2.      Materials and methods

Definitions of variables

-It is a little confusing when authors use the word “family” (i.e. family structure, main family occupations, and family income). I am not very clear what definitions of “family” in this study are. It seems that authors use the words “family” and “household” interchangeably. It would be helpful to include more detailed explanations on definitions of these variables. For example, does family structure refers to family members living in the same household? If so, the word “household structure” would suit better. How was “main family occupations” defined? Are they occupations of study participants’ parents or other guardians? How was “family income” defined? Please clarify.

-Line 162-163, please clarify if the availability of FV was measured by family’s ownership of farm which grows FVs only. If so, please explain why this particular measure was employed in this study.

Ethical considerations

-This study includes early adolescents (10-15 years old) who are often considered as minor populations. Their participation in the study may require consents from adolescents’ parents in Nepal. Please indicate if authors obtained legally valid parental consents.

3.      Results

Personal profiles of the study subjects

-It would be very helpful to see characteristics of the current study’s sample in comparison to the nationally representative sample of adolescents in Nepal in this result section, as authors discuss differences between the current study and other Nepalese studies at the beginning of Discussion section.

Discussion

-It is not very clear why authors claim that younger study subjects of the current study may have had greater access to FVs, fewer barriers to access to FV consumption and greater awareness of the benefits of eating FV than other populations in Nepal (line 229-230). Please discuss this with existing evidence.

-Line 233, authors indicate gender may play an important role to determine FV consumption. Please elaborate on how gender or cultural norms related to gender influence FV consumption based on existing evidence.

-Line 254, please provide examples of “robust study instruments” employed in this study.

-Although authors claim that this study aims to contribute to developing evidence-based interventions in Introduction (line 91-92), there is no discussion on this in Discussion section. It would be useful if the reader could see how the current findings fill the gaps in existing intervention and prevention programs in Nepal.

Author Response

Reviewer: third

Round: first

Manuscript ID: ijerph-505147

Manuscript title: Socio-demographic and diet-related factors associated with insufficient fruit and vegetable consumption among adolescent girls in rural communities of Southern Nepal

Comments and Suggestions for Authors

General comments: Dear Editor, Thank you for the opportunity to review the manuscript, Socio-economic and diet-related factors associated with insufficient fruit and vegetable consumption among adolescent girls in rural communities of Southern Nepal. The manuscript examines factors associated with FV consumptions among adolescent girls in rural Nepal, and has potentials to highlight key determinants of FV consumption in rural communities if authors could clarify the following parts.

Response: Thank you very much for good comments that helped to improve our manuscript in a right track. We have revised the manuscript in a number of spaces based on reviewers’ comments and the revised contents have been marked with blue colored writings to allow the reviewers’ verifications.

 1.      Introduction

Comment: -It would be helpful for the international audience to know why adolescent girls in Dhanusha district was selected to be an interesting case study to identify factors associated with FV consumption. For example, why only adolescent girls were selected? Why adolescent boys were not included? How is Dhausha district different from Singhuli district studied by Dhungana et al (2014)?

Response: Agree. We have now addressed this comment as one of the limitations of this study. Also, we have attempted to justify the rationale of choosing adolescent girls in the introduction section of this revised version of the manuscript. For your kind information, the current study was done in Southern Nepal (plain land-Terai), while the study performed by Dhungana et al (2014) was done in hilly region of Nepal.

Comment: -“STEPS survey” in line 71 should be defined in parenthesis the first time appeared in the main text.

Response: Agree. Revised.

2.      Materials and methods

Definitions of variables

Comment: -It is a little confusing when authors use the word “family” (i.e. family structure, main family occupations, and family income). I am not very clear what definitions of “family” in this study are. It seems that the authors use the words “family” and “household” interchangeably. It would be helpful to include more detailed explanations on definitions of these variables. For example, does family structure refers to family members living in the same household? If so, the word “household structure” would suit better. How was “main family occupations” defined? Are they occupations of study participants’ parents or other guardians? How was “family income” defined? Please clarify.

Response: Agree. Revised as suggested.

Comment: -Line 162-163, please clarify if the availability of FV was measured by family’s ownership of farm which grows FVs only. If so, please explain why this particular measure was employed in this study.

Response: Agree. Revised.

Ethical considerations

Comment: -This study includes early adolescents (10-15 years old) who are often considered as minor populations. Their participation in the study may require consents from adolescents’ parents in Nepal. Please indicate if authors obtained legally valid parental consents.

Response: Agree. We had received the parenteral consents for each study subjects who were below 18 years of age. The same has been mentioned in this revised version of the manuscript.

 3.      Results

Personal profiles of the study subjects

Comment: -It would be very helpful to see characteristics of the current study’s sample in comparison to the nationally representative sample of adolescents in Nepal in this result section, as authors discuss differences between the current study and other Nepalese studies at the beginning of Discussion section.

Response: Partially agree. Hope, information included about Nepalese adolescents from studies that were performed in nationally representative samples in the introduction and discussion section can address this issue.

 Discussion

Comment:-It is not very clear why authors claim that younger study subjects of the current study may have had greater access to FVs, fewer barriers to access to FV consumption and greater awareness of the benefits of eating FV than other populations in Nepal (line 229-230). Please discuss this with existing evidence.

Response: Agree. Revised.

Comment:-Line 233, authors indicate gender may play an important role to determine FV consumption. Please elaborate on how gender or cultural norms related to gender influence FV consumption based on existing evidence.

Response: Agree. Revised.

Comment:-Line 254, please provide examples of “robust study instruments” employed in this study.

Response: Agree. Revised.

Comment:-Although authors claim that this study aims to contribute to developing evidence-based interventions in Introduction (line 91-92), there is no discussion on this in the Discussion section. It would be useful if the reader could see how the current findings fill the gaps in existing intervention and prevention programs in Nepal.

Response: Agree. Thank you very much for excellent inputs given to us. We have addressed this comment by adding the study implications in the discussion section of this revised version of the manuscript.

Round 2

Reviewer 2 Report

The authors have addressed most of the comments on the first version. However, there are still some minor changes to be considered:

Line 84- 87: the ideas in this two sentence were not presented in a logical way. If the author want to emphasise the importance of addressing adolescent girls, they should move the “consumption of recommended amounts of FVs is dependent of many factors” to elsewhere.

Line 92 : suggest to rephrase as : identify “the associations of socio-demographic” and diet-related factors with insufficient FV consumption.

Line 126: anthropomorphic should be a typo

Line 127: It is still unclear whether the fruit and vegetable consumption here means FFQ? If they are the same, please combine

Line 147: fruit and vegetable consumption should be the dependent variable, not independent variable

Table 2: Awareness regarding fruit and vegetable “consumption”, the word consumption is missing

Frequency of food should be frequency of “meal consumption per day”

Add “once per week” after processed food consumption

Line 218 to 225: suggest to revise as follow:

Education level, household income, unaware awareness of recommended FVs intake, non-availability of FVs, dietary diversity and body mass index were significantly associated variables determinants with of insufficient FV consumption. Adolescent girls who were educated with <10th grade of schooling were more likely (adjusted odds ratio (aOR) 2.5; 95% CI (confidence interval) (1.06-5.78)) to have been consumed insufficient fruit and vegetable than their counterparts of ≥10th grade of schooling. Similarly, household income in the first tercile (aOR 3.9; 95% CI (1.22-12.67)), those who were unaware regarding of the recommended fruit and vegetable consumption (aOR 3.1; 95% CI (1.20-8.23)), non-availability of FVs at household level (aOR 3.0; 95% CI  (1.16-7.60))

Line 236: suggest to revise as follow: This finding is was higher as compared with some than those reported in other

Line 242-243: suggest to revise as follow:

might have to dependent more on household foods availed at households than older age groups who might have preferred of preference on packed pre-packaged and fast foods usually outside households leading to decreased FV consumption

Line 248: suggest to revise as follow:

Other studies [45,46] reported that compared with girls,

Author Response

Reviewer: Second

Round: Second

Revision type: Minor

Manuscript ID: ijerph-505147

Manuscript title: Socio-demographic and diet-related factors associated with insufficient fruit and vegetable consumption among adolescent girls in rural communities of Southern Nepal

Comments and Suggestions for Authors

General comments: The authors have addressed most of the comments on the first version. However, there are still some minor changes to be considered:

Response: Thank you very much for your encouragement and appreciation. We highly appreciate reviewer’s every comments that helped us a lot to improve our manuscript. We have revised the manuscript in a number of spaces based on reviewers’ comments and the second revised contents have been marked with green colored writings to allow the reviewers’ verifications.

Comment: Line 84- 87: the ideas in this two sentence were not presented in a logical way. If the author want to emphasise the importance of addressing adolescent girls, they should move the “consumption of recommended amounts of FVs is dependent of many factors” to elsewhere.

Response: Agree. revised

Comment: Line 92 : suggest to rephrase as : identify “the associations of socio-demographic” and diet-related factors with insufficient FV consumption.

Response: Agree. Corrected.

Comment: Line 126: anthropomorphic should be a typo

Response: Agree. Corrected.

Comment: Line 127: It is still unclear whether the fruit and vegetable consumption here means FFQ? If they are the same, please combine

Response:  It is about where the study questionnaires were adapted from. Therefore, it has been revised as: Nepalese Adolescent Nutrition Survey (2014), the Food Frequency Questionnaire (FFQ), and the dietary diversity questionnaire [38-40]; all of which were of already piloted study instruments.

Comment: Line 147: fruit and vegetable consumption should be the dependent variable, not independent variable

Response: Agree. Corrected.

Comment: Table 2: Awareness regarding fruit and vegetable “consumption”, the word consumption is missing

Response: Agree. Revised

Comment: Frequency of food should be frequency of “meal consumption per day”

Response: Agree. Revised

Comment: Add “once per week” after processed food consumption

Response: Agree. Revised

 Comment: Line 218 to 225: suggest to revise as follow:

 Education level, household income, unaware awareness of recommended FVs intake, non-availability of FVs, dietary diversity and body mass index were significantly associated variablesdeterminants with of insufficient FV consumption. Adolescent girls who were educated with <10th grade of schooling were more likely (adjusted odds ratio (aOR) 2.5; 95% CI (confidence interval) (1.06-5.78)) to have been consumed insufficient fruit and vegetable than their counterparts of ≥10th grade of schooling. Similarly, household income in the first tercile (aOR 3.9; 95% CI (1.22-12.67)), those who were unaware regarding of the recommended fruit and vegetable consumption (aOR 3.1; 95% CI (1.20-8.23)), non-availability of FVs at household level (aOR 3.0; 95% CI  (1.16-7.60))

Response: Agree. Revised as suggested.

Comment:  Line 236: suggest to revise as follow: This finding is was higher as compared with some than those reported in other

Response: Agree. Revised.

Comment:  Line 242-243: suggest to revise as follow:

might have to dependent more on household foods availed at households than older age groups who might have preferred of preference on packed pre-packaged and fast foods usually outside households leading to decreased FV consumption

Response: Agree. Revised as suggested.

 Comment: Line 248: suggest to revise as follow:

Other studies [45,46] reported that compared with girls,

Response: Agree. Revised.

Reviewer 3 Report

Dear Editor,

I am happy with most of the revision. I suggest the authors to copy edit the revised parts. There are typos and grammatical errors. 

Author Response

Comments and Suggestions for Authors

Reviewer: Third

Round: Second

Revision type: Minor

Manuscript ID: ijerph-505147

Manuscript title: Socio-demographic and diet-related factors associated with insufficient fruit and vegetable consumption among adolescent girls in rural communities of Southern Nepal

Comment: Dear Editor, I am happy with most of the revision. I suggest the authors to copy edit the revised parts. There are typos and grammatical errors. 

Response: Thank you very much for the encouragement and appreciation. We highly appreciate reviewer’s every comments that helped us a lot to improve our manuscript. We have revised the manuscript in a number of spaces based on reviewers’ comments and the second revised contents have been marked with green colored writings to allow the reviewers’ verifications. Further, we have checked typos and grammatical errors and get corrected.